# Mapping Forest Fire Risk and Development of Early Warning System for NW Vietnam Using AHP and MCA/GIS Methods

**Thanh Van Hoang [1], Tien Yin Chou [1], Yao Min Fang [1], Ngoc Thach Nguyen [2], Quoc Huy Nguyen [2,3,*], Pham Xuan Canh [2], Dang Ngo Bao Toan [4], Xuan Linh Nguyen [2] and Michael E. Meadows [5,6,7]**

[1] Geographic Information Systems Research Center, Feng Chia University, Taichung 40724, Taiwan; van@gis.tw (T.V.H.); jimmy@gis.tw (T.Y.C.); frankfang@gis.tw (Y.M.F.)

[2] Faculty of Geography, VNU University of Science, No 334 Nguyen Trai, Thanh Xuan, Hanoi City 100000, Vietnam; nguyenngocthachhus@gmail.com (N.T.N.); xuancanhhus@gmail.com (P.X.C.); thayninh@gmail.com (X.L.N.)

[3] Ph.D. Program for Civil Engineering, Water Resources Engineering, and Infrastructure Planning, College of Construction and Development, Feng Chia University, Taichung 40724, Taiwan

[4] Faculty of Cadastral and Geography, Quy Nhon University, 170 An Duong Vuong, Nguyen Van Cu, Quy Nhon, Binh Dinh 55113, Vietnam; dnbtoan@qnu.edu.vn

[5] Department of Environmental & Geographical Science, University of Cape Town, Rondebosch 7701, South Africa; michael.meadows@uct.ac.za

[6] School of Geographic Sciences, East China Normal University, Shanghai 200241, China

[7] College of Geography and Environmental Sciences, Zhejiang Normal University, Jinhua 321004, China

**\*** Correspondence: st_huy@gis.tw

**Abstract:** Forest fires constitute a major environmental problem in tropical countries, especially in the context of climate change and increasing human populations. This paper aims to identify the causes of frequent forest fires in Son La Province, a fire-prone and forested mountainous region in northwest Vietnam, with a view to constructing a forest fire-related database with multiple layers of natural, social and economic information, extracted largely on the basis of Landsat 7 images. The assessment followed an expert systems approach, applying multi-criteria analysis (MCA) with an analytical hierarchy process (AHP) to determine the weights of the individual parameters related to forest fires. A multi-indicator function with nine parameters was constructed to establish a forest fire risk map at a scale of 1:100,000 for use at the provincial level. The results were verified through regression analysis, yielding $R^2 = 0.86$. A real-time early warning system for forest fire areas has been developed for practical use by the relevant government authorities to provide more effective forest fire prevention planning for Son La Province.

**Keywords:** forest fires; multi-criteria; analytic hierarchy process; the fire triangle; disaster mitigation; remote sensing

## 1. Introduction

Vietnam is a developing world country in the tropics which is rich in biodiversity, but especially vulnerable to environmental deterioration through a combination of the exploitation of its natural resources against the background of an increasing population, climate change and sea level rise [1,2]. Tropical evergreen and deciduous forests occupy more than one third of the landscape in Vietnam and the degradation of these natural forests, in addition to its plantations, is recognized as a significant

environmental problem [3]. A range of factors, including greater frequency and a magnitude of drought and increased air temperatures, have been attributed to the increased frequency of wildfires in Vietnam [4]. Forest fires cause severe damage in both environmental and socioeconomic terms and threaten human lives and livelihoods [5]. In recent years in Vietnam, major forest fires have resulted in serious economic and environmental damage and contributed to increasing environmental pollution [3]. According to statistics from the Ministry of Agriculture and Rural Development, as of December 2016, there were almost 400 forest fires in 2015 [6] and it is therefore, a pressing issue for each forested area to take appropriate measures to accurately map the fire risk, and to develop early warning and forest fire prevention plans.

Forest fires are a product of interactions between several environmental factors, including fuel availability, weather, topography and source of ignition. When factors such as low humidity, strong wind, topography and wind direction are favorable, a fire can develop rapidly if the quantity and availability of fuel is appropriate [7–9]. Forest fire risk is a term used to describe the possibility of forest fires [10], and is usually classified into different levels of risk. The concept of forest fire risk (and the risk of any hazard, for that matter) is not just about the probability of an event, but also about the consequences of the event. The prediction of forest fire risk prediction involves the identification of fire risk levels which are strongly influenced by weather conditions and other factors including the condition of the vegetation and the topography. For example, the hotter, drier and longer the weather conditions are, the greater the risk of forest fires. The risk of forest fires also depends on vegetation characteristics. For example, forests with many oil trees, vines, or branches with dry leaves are more prone to fire. Accordingly, fire risk assessment is based on the results of an analysis of weather characteristics and forest conditions [11–13].

## 1.1. Environmental Factors Affecting Forest Fire

### 1.1.1. Terrain

Terrain, or topography, comprises slope angle, aspect, and elevation. Terrain complexity (texture) may in turn affect changes in the fuel and atmospheric conditions [14]. In the northern hemisphere, south and south-west slope directions are considered to be more fire prone as they receive more sunlight and have lower humidity and higher fuel temperatures. Slope angles of 15°–20° have been shown to be especially favorable to fire spread [9]. Elevation also influences fire behavior and response. Higher elevations in the region are associated with lower humidity and increased wind speeds. In addition, the rugged topography at higher elevations and remoteness make it more difficult to respond to any forest fire, leading to an increased risk of wildfire.

### 1.1.2. Combustible Fuel

Fuel is an important element of the fire triangle (Figure 1), affecting the flammability as well as the size and intensity of a fire. Fuel is described both in terms of fuel state and fuel type. The fuel state refers to the moisture content of the fuel, whether a plant is alive or dead. The fuel type includes the physical characteristics of the fuel, the composition of the fuel and the fuel group. The physical nature of the fuel affects how the fuel burns, including the number, size, alignment and arrangement of the material [10,15,16]. The most important effect of moisture on fire is the effect of steam released from burning fuel, since this reduces the amount of available oxygen and suppresses combustion.

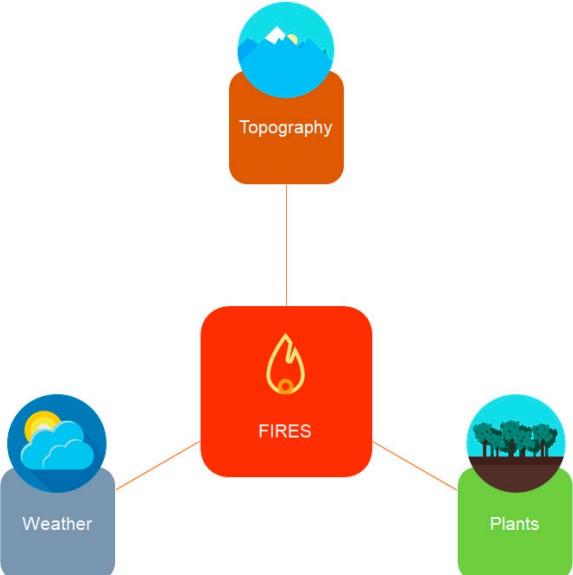

**Figure 1.** The 'fire triangle': fire risk assessment must account for the characteristics of all three elements. This representation illustrates a situation without human influence, which also affects fire risk.

### 1.1.3. Weather Conditions

The occurrence of fire is very closely related to weather, both prevailing and antecedent [15,17]. The fire risk index (FRI) [7,8] is strongly determined by air temperature, relative humidity and wind speed. For example, elevated air temperatures reduce the amount of heat needed to ignite materials and also lowers humidity, while higher wind speeds lead to accelerated fire spread and greater fire intensity [9,15].

### 1.1.4. Human Impact

Most fires in Vietnam are either directly or indirectly related to human activities, [3,18] and, indeed, many forest areas are adjacent to urban settlements or are in agricultural production areas where slash and burn cultivation is less regulated, which clearly increases the risk of wildfires. It has been reported [19] that anthropogenic causes include the burning of agricultural land, the hunting of wild animals, the harvesting of honey using smoke, and even trading conflicts.

### 1.2. Aim, Objectives and Data Sources

The aim of this research was to generate a fire risk map on the scale of 1:100,000 for Son La Province and to develop a real-time early warning system. This was achieved through the application of multi-criteria analysis (MCA) and GIS tools to analyze the relationship between natural and social factors and the risk of forest fire. The data sources included the following: (1) 1:100,000 topographic background dataset from the Ministry of Natural Resources and Environment; (2) data on forest cover, population and the current status of resource exploitation from the Department of Agriculture and Rural Development, Department of Natural Resources and Environment of Son La Province, edited and supplemented by Landsat imagery; (3) the meteorological data extracted from the Central Meteorological Office of Vietnam supplemented by local station records (seven stations). In order to make more practical use of the forest fire risk map, we also developed an early warning system that the authorities can use to obtain information in advance that forest fire risk is high in a particular locality.

## 2. Study Area

Son La is a mountainous province in northwest Vietnam, with an area of 14,125 km$^2$, accounting for 4.27% of the total area of Vietnam. Ranking third among 63 provinces and cities [20] and is typical of the

forested areas of the country. Son La lies between 20°39′–22°02′ N l and 103°11′–105°02′ E. bordering Yen Bai, Dien Bien, and Lai Chau provinces to the north, Phu Tho and Hoa Binh provinces to the east, Dien Bien province to the west, Thanh Hoa and Huaphanh (Laos) to the south, and Luangprabang Province (Laos) to the northwest (Figure 2). In 2018, Son La had approximately 1.24 million people with a population density of 88 people/km² [20]. Son La is located at a mean elevation of 600–700 m above sea level; the highest mountain is Pha Luong, with an elevation of nearly 2000 m. The terrain is highly fragmented, and 97% of the natural area is in the Da River and Ma River basins. There are two plateaus, Moc Chau and Son La, with relatively flat terrain.

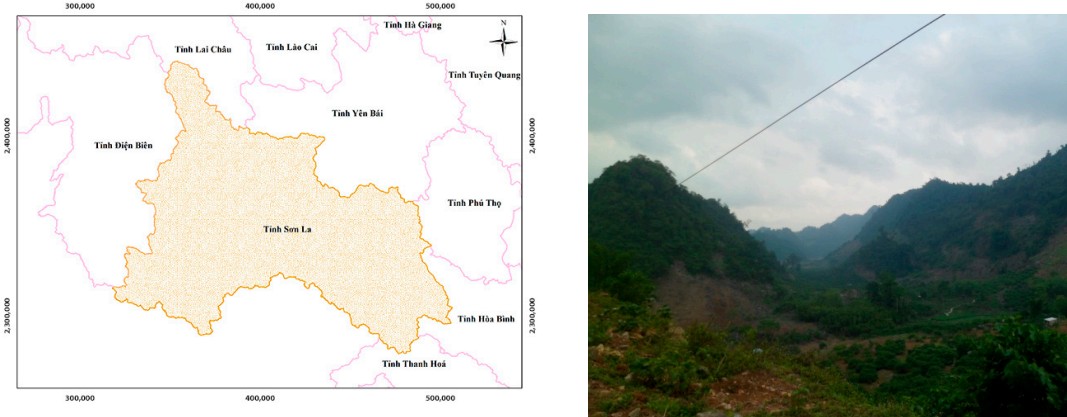

**Figure 2.** Study area: Son La Province and its typical landscape.

The climate is humid subtropical, although there is a cooler and drier season in the winter that favors fire [21]. In the dry season, the weather is periodically hot and the adiabatic west wind lowers humidity and dries out foliage, thereby increasing flammability. The forests of Son La are diverse, accounting for 73% of the total area of the province covering some 357.000 ha [21] supplemented by restored forest and agro-forestry (Table 1). Plantation forest is mainly pine, with acacia species grown in the lowlands.

**Table 1.** Land use in Son La Province [20].

| ID | Land Use | Area (ha) | Area (%) |
|----|----------|-----------|----------|
| 1 | Residential | 4873 | 0.29 |
| 2 | Agricultural land, other land | 390,917.70 | 22.88 |
| 3 | Vacant land | 605,797 | 35.46 |
| 4 | Water surface | 25,728.40 | 1.51 |
| 5 | Rocky mountains without trees | 7147.6 | 0.4 |
| 6 | Mixed bamboo, wood forest | 16,928.30 | 0.99 |
| 7 | Rich evergreen broad leaved forest | 11,023 | 0.65 |
| 8 | Poor evergreen broad leaved forest | 62,970.80 | 3.69 |
| 9 | Restored evergreen broad leaved forest | 368,903.20 | 21.59 |
| 10 | Medium evergreen broad leaved forest | 46,886.70 | 2.74 |
| 11 | Rocky mountain forest | 91,500.30 | 5.36 |
| 12 | Bamboo forest | 56,748 | 3.32 |
| 13 | Planted forest | 19,147.90 | 1.12 |
| | **Total** | **1,708,571.90** | **100.00** |

## 3. Methods

### 3.1. Satellite Remote Sensing Mapping and Multi-Criteria Analysis (MCA)

According to available statistics, between 2014 and 2018, there were 327 fires in Son La [21]. Forest fires were mostly concentrated in the Muong La, Son La and Sop Cop districts of the province

(Figure 3). Satellite imagery was used to identify the hotspots in the thermal infrared channel and to construct a forest fire database through image interpretation and classification. Geospatial technology, including several GIS software packages, such as ArcGIS, QGIS, and multi-criteria spatial analysis in GIS (multi-criteria analysis—MCA), and AHP (the analytic hierarchy process) [22] were applied to mapping and integrating the information related to the forest fires. This enabled the identification of the causes and the system interactions between the forest fire parameters, the calculation of the weights of each parameter, and the integration into a general model to develop a forest fire risk map. The forest fire risk, classified into five levels, was calculated using the following multi-criteria integration function:

$$FR = \sum_{i=1}^{n} (w_i x_i) \qquad (1)$$

where:

FR is the risk of forest fire;

$w_i$: weight of indicator (i);

$x_i$: indicator (i);

n: number of indicators (from 1 to n).

Depending on the conditions specific to each geographic area and on the availability of data, it was not always possible to apply exactly the same evaluation criteria and selected parameters, although this results in only minor disparities.

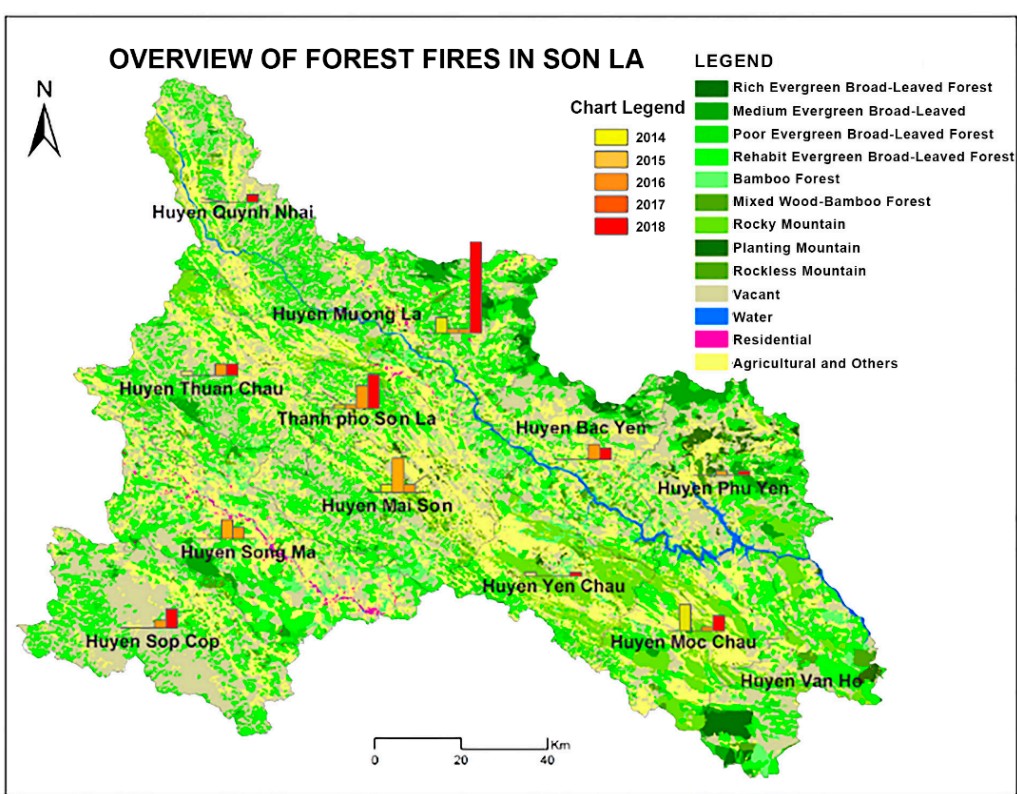

**Figure 3.** Map of the forest fires in Son La (2014–2018).

## 3.2. Development of the Forest Fire Risk Map

In assessing the risk of forest fire, many sources of information can be used, including satellite imagery, land use maps, administrative boundary information, forest distribution and statistical information on economic and social characteristics. In addition, the application of any risk assessment is dependent on its suitability for stakeholders, including land use policy makers and land users themselves such that it becomes a multi-criteria decision-making issue. The MCA method is therefore

useful in classifying and weighting the various criteria and has been widely applied [23–25]. MCA steps include criteria identification, quantification and analysis for the evaluator and the combination of judgments [24]. Among the many methods of MCA, combining linear weights with spatial attributes using overlays is most often used because of its simplicity. The analytical hierarchy process (AHP) method is also frequently applied to forest fire risk assessment [26,27] and has the advantage of breaking down the problem into a hierarchical structure, allowing the involvement of experts and stakeholders in the evaluation. Here, the weightings of indicators were estimated separately using AHP to establish a comparison matrix via questionnaires. The MCA with AHP technique determines the weight of the factors and synthesizes the knowledge of appropriate experts in diverse fields, especially those familiar with the study area. Mapping forest fire risks is developed from integrated decisions [17], and GIS technology is used for spatial analysis in constructing the database, analyzing and assessing hotspot information, and in the spatial representation of possible adaptation and mitigation measures.

### 3.3. Data Integration

The procedure for integrating the various data sources and establishing a forest fire risk map is shown in Figure 4. In this procedure, the parameters are integrated into a model incorporating both natural and social classes of information. Natural classes include the following: the forest type (forest map 1:100,000) established by remote sensing methods, elevation, topographic slope, slope direction, river density (analyzed from the terrain map 1:100,000), total annual precipitation, maximum temperature/year, minimum humidity/year, and the prevailing wind direction (extracted from 33-year meteorological data). Social information consists of the population distribution and the distance to roads, sourced from provincial statistics.

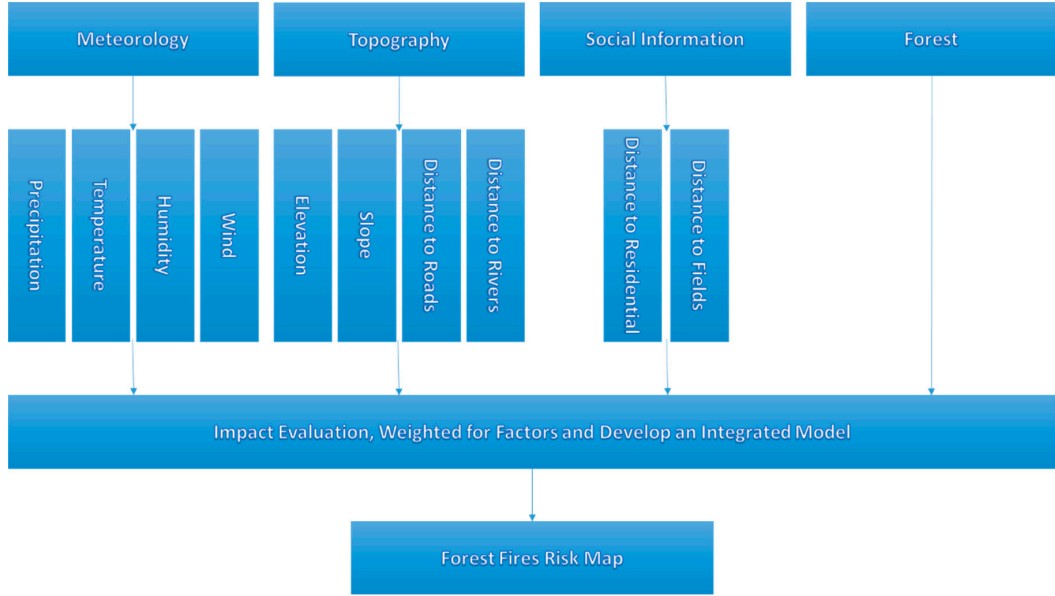

**Figure 4.** Process of developing a forest fire risk map.

### 3.4. Creation of a Database for Forest Fires

The data were processed by ArcGIS software and standardized according to ISO TC211 at a scale of 1:100,000 (Figure 5).

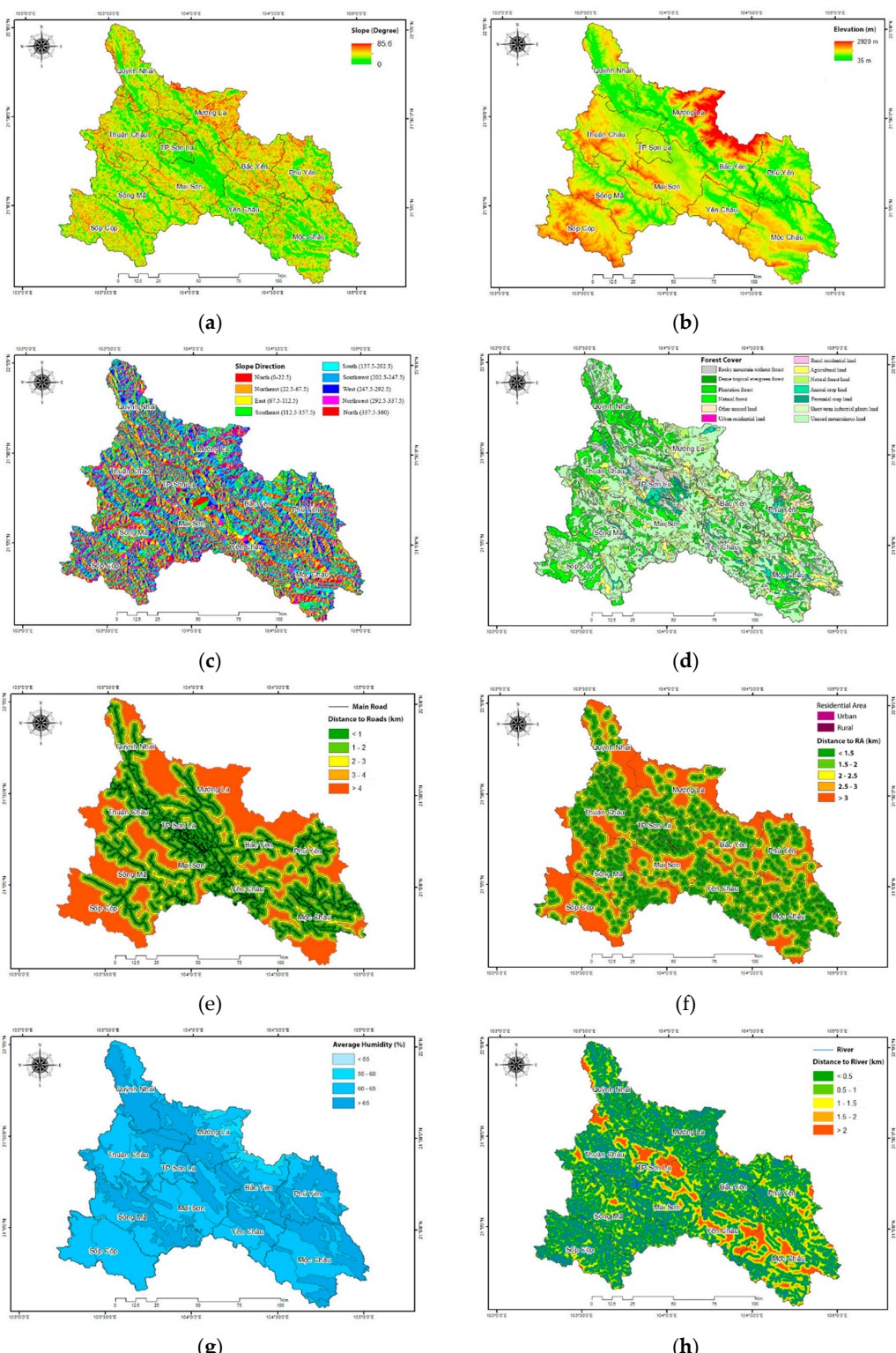

**Figure 5.** *Cont.*

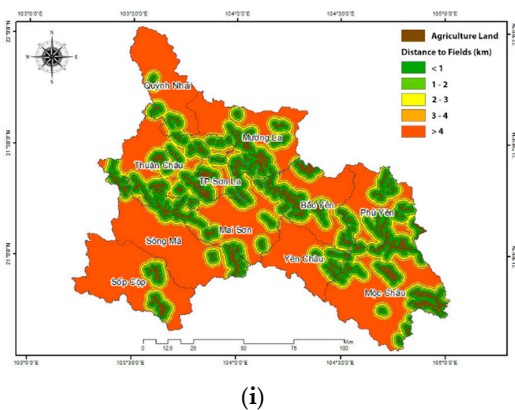

(**i**)

**Figure 5.** Information layers for the establishment of a forest fire risk map in Son La Province: (**a**) slope; (**b**) elevation; (**c**) slope direction; (**d**) forest cover; (**e**) distance to roads; (**f**) distance to residential areas; (**g**) humidity; (**h**) distance to rivers; and (**i**) distance to fields.

Information layers include:

- Digital elevation model (DEM), slope and gradient data extracted from the 1:100,000 topographic map. Forest data collected from the provincial forest management department, corrected and supplemented with the Landsat satellite image.
- Population distribution and distance to residential areas. Distance to roads, data on rivers and streams and the density of rivers and streams in the study area.
- Current status of upland fields and distance to upland fields (currently, the slash and burn form of agriculture in the upland field area still exists in the mountainous regions of Vietnam. Every year, people adopt the slash and burn method before planting a new crop without following any specific procedure or law. This causes many forest fires. Forest areas closer to the slash and burn agricultural areas have a higher potential for forest fires. Since these field locations are usually far from the villages, the distance to the upland fields is one of the main factors to assess forest fire risk.
- Data on monthly average humidity, based on the meteorological data of the study area (33-years of statistic data from the Central Meteorological and Hydrological Station).

*3.5. Evaluation of Information Layers for Assessment of Forest Fire Risk*

A five-point evaluation scale was conducted independently for each of the nine information layers whereby a score of 1 indicates a low risk, through to a score of 5 which indicates extreme risk, as shown in Table 2. After GIS processing, nine new data layers were collated, where each data layer was assigned a value from 1 to 5, indicating the degree of forest fire risk (Figure 6).

**Table 2.** Fire risk assessment for the forest information layers.

| Assessment Level | Slope (Degree) | Elevation (m) | Distance to Roads (km) | Slope Direction | Distance to River/Stream (m) | Humidity (%) | Distance to Residential Areas (km) | Distance to Fields (m) |
|---|---|---|---|---|---|---|---|---|
| 1—Very low | 0°–8° | <1100 | >2 | North Southeast | <200 | 60–100 | <1.5 | >400 |
| 2—Low | 8°–15° | 1100–1200 | 1.5–2 | Northeast | 200–400 | 55–60 | 1.5–2 | 350–400 |
| 3—Medium | 15°–25° | 1200–1300 | 1–1.5 | Southeast | 400–600 | 50–55 | 2–2.5 | 300–350 |
| 4—High | 25°–45° | 1300–1600 | 0.5–1 | Southwest Northwest | 600–800 | 45–50 | 2.5–3 | 200–300 |
| 5—Very high | >45° | >1600 | <0.5 | West | >800 | <45 | >3 | <200 |

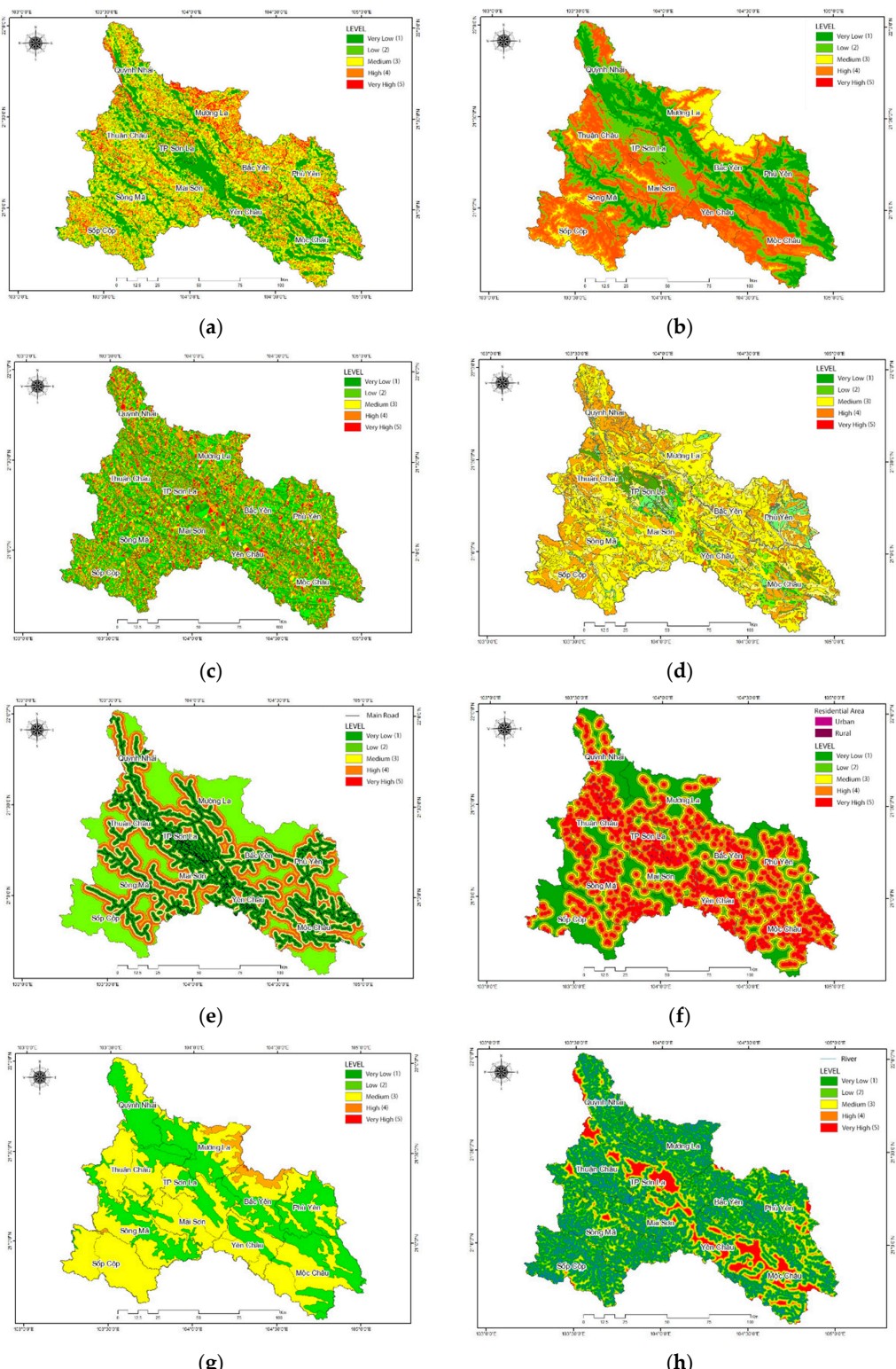

**Figure 6.** *Cont.*

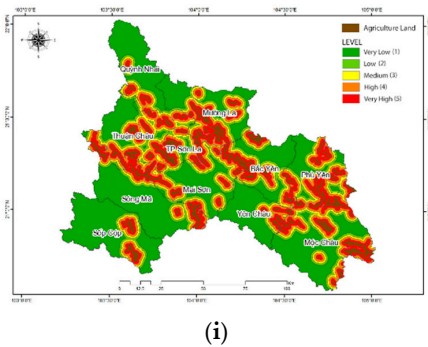

(**i**)

**Figure 6.** Evaluative layers for establishing a forest fire risk map in Son La Province: (**a**) slope; (**b**) elevation; (**c**) slope direction; (**d**) forest cover; (**e**) distance to roads; (**f**) distance to residential areas; (**g**) humidity; (**h**) distance to rivers; and (**i**) distance to fields.

*3.6. Determining the Weights of the Components Contributing to Fire Risk*

To determine the weights of the components, this study used a pairwise comparative analysis (AHP), which is a method applied to multi-objective decision problems [28]. In this study, the opinions of many experts [6,11,29,30] were referenced to construct a comparison matrix for nine classes of information for the construction of the forest fire risk map relating to: forest cover (FC), humidity (HM), elevation (EL), distance to roads (DT), slope (SL), stream density (SD), slope direction (SI), distance to fields (UF), and the distance to residential areas (RA) (Table 3).

**Table 3.** Comparison matrix and weighted value of indicators based on the analytical hierarchy process (AHP) method.

| Indicator | FC | HM | EL | DT | SL | SD | SI | UF | RA | Weight |
|:---:|:---:|:---:|:---:|:---:|:---:|:---:|:---:|:---:|:---:|:---:|
| FC | 0.26 | 0.23 | 0.28 | 0.34 | 0.26 | 0.23 | 0.20 | 0.18 | 0.16 | 0.24 |
| HM | 0.26 | 0.23 | 0.28 | 0.23 | 0.20 | 0.18 | 0.16 | 0.14 | 0.16 | 0.20 |
| EL | 0.13 | 0.12 | 0.14 | 0.11 | 0.26 | 0.23 | 0.20 | 0.11 | 0.14 | 0.16 |
| DT | 0.09 | 0.12 | 0.14 | 0.11 | 0.13 | 0.12 | 0.12 | 0.14 | 0.11 | 0.12 |
| SL | 0.06 | 0.08 | 0.03 | 0.06 | 0.07 | 0.12 | 0.16 | 0.14 | 0.14 | 0.09 |
| SD | 0.06 | 0.08 | 0.03 | 0.06 | 0.03 | 0.06 | 0.08 | 0.11 | 0.11 | 0.07 |
| SI | 0.05 | 0.06 | 0.03 | 0.04 | 0.02 | 0.03 | 0.04 | 0.14 | 0.11 | 0.06 |
| UF | 0.05 | 0.06 | 0.05 | 0.03 | 0.02 | 0.02 | 0.01 | 0.04 | 0.05 | 0.04 |
| RA | 0.04 | 0.04 | 0.03 | 0.03 | 0.01 | 0.01 | 0.01 | 0.02 | 0.03 | 0.02 |

## 4. Results

*4.1. Fire Risk Mapping and Integration Equation*

Results are shown to have a consistent coefficient, CR = CI/RI = 0.062 < 0.1, ensuring the reliability in computation [31], where CR is a consistency ratio, CI is the random coincidence index, and the RI is the random number determined in n (number of factors). With the weighting as determined in Table 4, the fire risk function equation incorporating all nine parameters is as follows:

$$CR = FC * 0.24 + HM * 0.20 + EL * 0.16 + DT * 0.12 + SL * 0.09 + SD * 0.07$$
$$+ SI * 0.06 + UF * 0.04 + RA * 0.02$$

(2)

**Table 4.** Comparison of the forest fires between 2011 and 2015 in Son La Province and the forest fire risk map.

| No. | District | 2011 | 2012 | 2013 | 2014 | 2015 | 2015 $\sum$ 5 Years 2011 | Level IV, V Risk Region Area (ha) |
|---|---|---|---|---|---|---|---|---|
| 1 | Moc Chau | 0 | 1 | 1 | 1 | 8 | 11 | 12,342.4 |
| 2 | Van Ho | 0 | 0 | 1 | 1 | 10 | 12 | 20,237.7 |
| 3 | Bac Yen | 2 | 1 | 3 | 2 | 16 | 24 | 31,793.2 |
| 4 | Ma River | 0 | 1 | 1 | 1 | 4 | 7 | 8310.3 |
| 5 | Quỳnh Nhai | 1 | 0 | 1 | 2 | 8 | 12 | 21,423.5 |
| 6 | Phu Yen | 1 | 0 | 0 | 2 | 2 | 5 | 10,589.8 |
| 7 | Yen Chau | 0 | 2 | 1 | 1 | 3 | 7 | 15,604.9 |
| 8 | Thuan Chau | 2 | 3 | 0 | 2 | 10 | 16 | 28,339.2 |
| 9 | Son La city | 1 | 1 | 0 | 2 | 3 | 7 | 8879.5 |
| 10 | Soc Cop | 1 | 1 | 2 | 3 | 8 | 15 | 16,699.2 |
| 11 | Muong La | 0 | 1 | 5 | 3 | 6 | 15 | 23,165.2 |
| 12 | Mai Son | 2 | 3 | 0 | 1 | 5 | 11 | 24,833.8 |

Results are classified into five levels, corresponding to low, medium, high, dangerous and extremely high fire risk (Figure 7a,b).

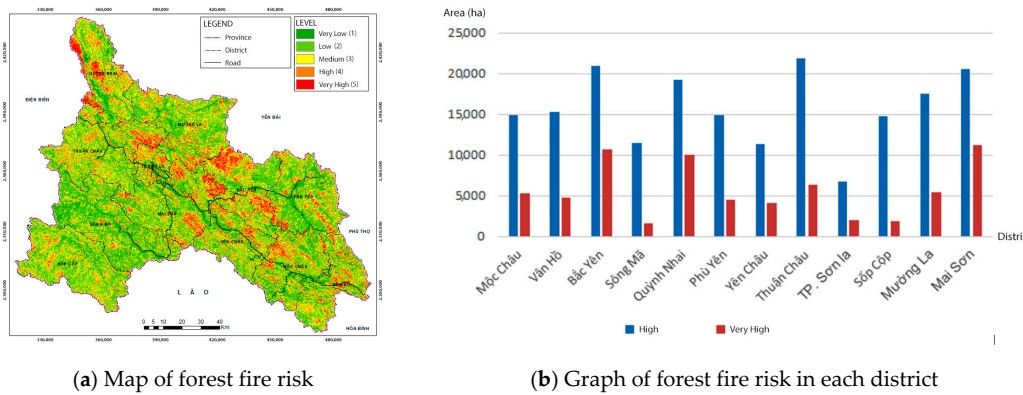

(**a**) Map of forest fire risk           (**b**) Graph of forest fire risk in each district

**Figure 7.** (**a**) Map of the forest fire risk in Son La province and (**b**) the fire risk plotted per district.

*4.2. Evaluation of Model Reliability*

To assess the reliability of the resulting map, we obtained and compiled an inventory of all the forest fires between 2014 and 2018 in each district (Table 5) and assessed the correlation between the total fire points and the totals of level 4 and 5 forest fire risks, for each district. Verification (Figure 8) indicates that the forest fire hazard map accurately reflects the current situation of areas at risk of fire in the forest areas. The results show that there is a correlation coefficient ($R^2 = 0.86$) between the forecasted forest fire risk area and the real forest fire area at risk levels 4 and 5 (Figure 9).

**Table 5.** Historical forest fires from 2014 to 2018 in Son La Province.

| No. | District | 2014 | 2015 | 2016 | 2017 | 2018 | 2018 $\sum$ 5 Years 2014 |
|-----|----------|------|------|------|------|------|--------------------------|
| 1 | Moc Chau | 0 | 1 | 1 | 1 | 8 | 11 |
| 2 | Van Ho | 0 | 0 | 1 | 1 | 10 | 12 |
| 3 | Bac Yen | 2 | 1 | 3 | 2 | 16 | 24 |
| 4 | Ma River | 0 | 1 | 1 | 1 | 4 | 7 |
| 5 | Quỳnh Nhai | 1 | 0 | 1 | 2 | 8 | 12 |
| 6 | Phu Yen | 1 | 0 | 0 | 2 | 2 | 5 |
| 7 | Yen Chau | 0 | 2 | 1 | 1 | 3 | 7 |
| 8 | Thuan Chau | 2 | 3 | 0 | 2 | 10 | 16 |
| 9 | Son La city | 1 | 1 | 0 | 2 | 3 | 7 |
| 10 | Soc Cop | 1 | 1 | 2 | 3 | 8 | 15 |
| 11 | Muong La | 0 | 1 | 5 | 3 | 6 | 15 |
| 12 | Mai Son | 2 | 3 | 0 | 1 | 5 | 11 |

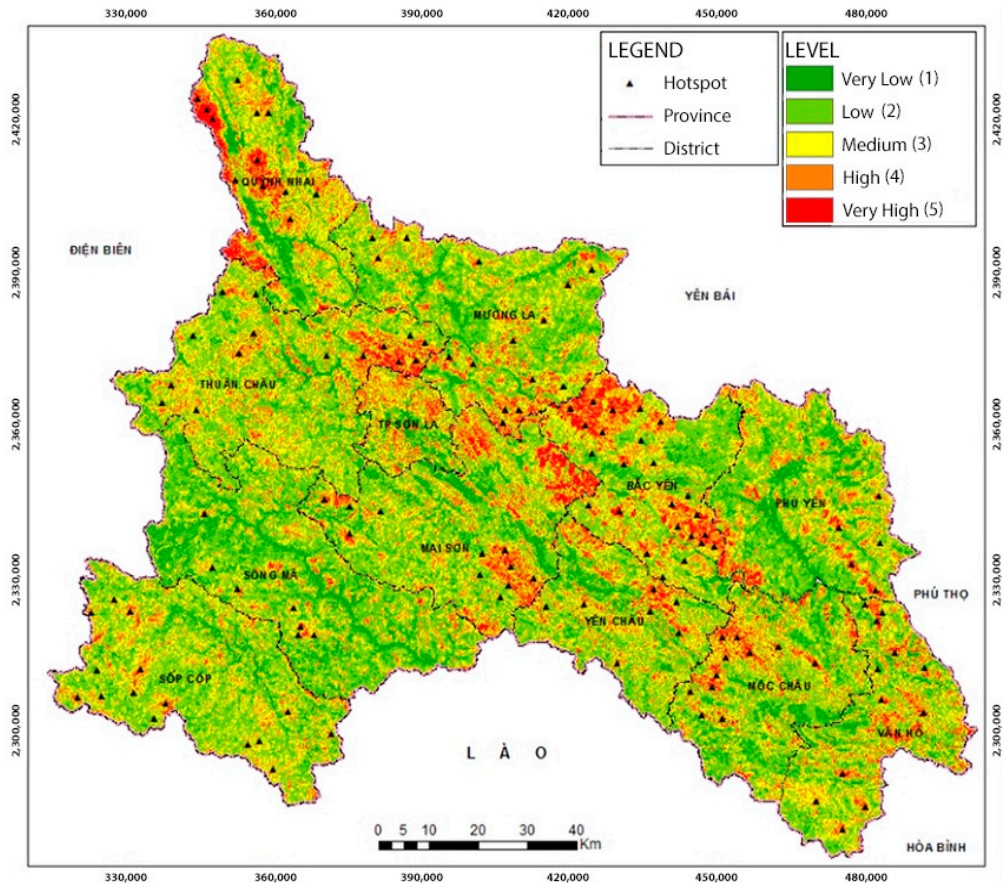

**Figure 8.** Map of fires in Son La Province from 2014 to 2018, based on forest fire risk maps. (Source: Son La Province).

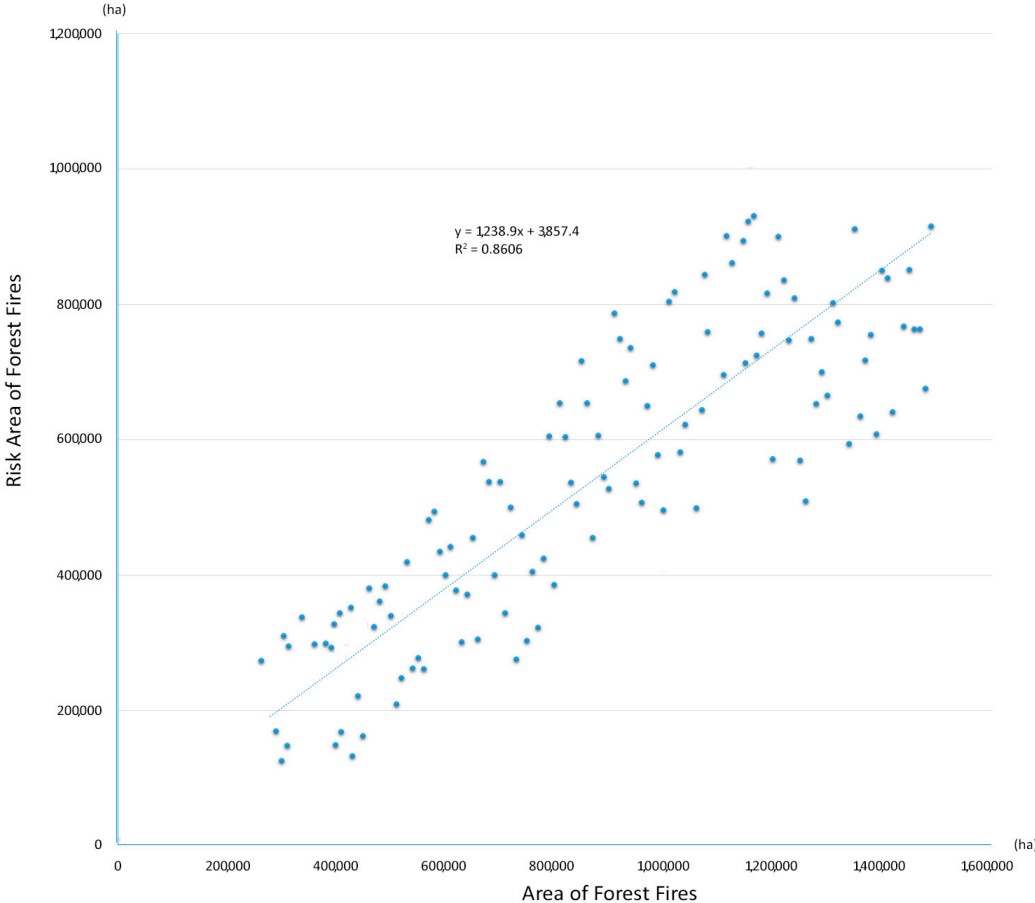

**Figure 9.** Correlation between the forecasted forest fire risk area and the actual forest fire area.

### 4.3. Early Warning System

Early Warning Systems [32–35] are in part social processes with various levels of complexity but require the collection and collation of accurate real-time data on ground conditions. The system structure developed here consists of two main parts, hardware and software. The hardware is an iMETOS IMT280 monitoring station which has the function of monitoring, through continuous transmission, all environmental conditions, namely the temperature, humidity, wind speed, wind direction and the visual imagery at locations with a high risk of forest fire. Each station has a rain gauge, and all the sensors enable the calculation of evapotranspiration by visualizing with the air temperature and relative humidity sensor, global radiation sensor, anemometer, circuit board control and communication equipment (USB 3G) and an internet protocol (IP) camera with 20Mp image resolution, camera holder and rotary motor, circuit board to control the camera's rotation mode, NUC PC computer to run image analysis and hte processing software and 3G USB to transmit fire information via the internet. Change in the values of smoke and fire obtained from the image provides the basis for reporting the characteristics of the fire generated. Images are taken continuously at 20 frames per second.

### 4.3.1. Software Database

The software is programmed on a computer that controls the operation of iMETOS stations and processes information from images, reports and transmits if a forest fire occurs. The software database includes: (1) the detailed information about the potential fire risks including: the coordinates, lot number, plot, sub-zone, name of forest owner; (2) the photographs of the current fire taken from the monitoring station; (3) the information about the fire transferred to the website for the administrators to confirm the information and transmit the information to manage the fire. A model of the overall system is shown in Figure 10.

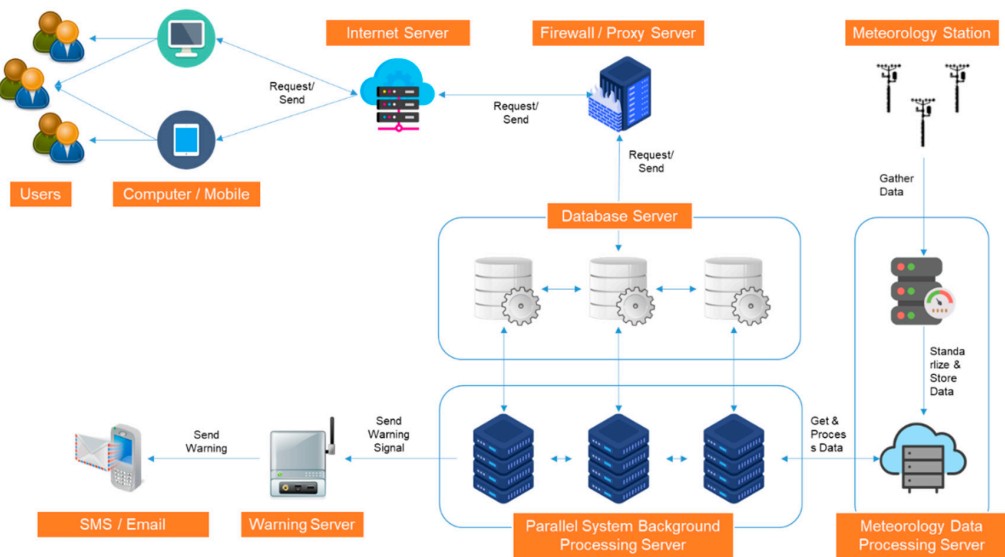

**Figure 10.** Model of the forest fire early warning system.

### 4.3.2. Software Function

The software uses web-based GIS tools with open source programming language Python, PHP (Personal Home Page) and a PostgreSQL/PostGIS database that is built into the server and integrated into the website, so that the administrators can manage the system via the internet. There are three functional software modules as follows: (1) the system administration component that controls the operation of the stations and the system user information through the internet; (2) the environmental information-processing component, displaying the temperature, wind direction, and humidity, and the information-processing operations for the administrators. In the event of a forest fire, after checking the actual information from the image of the monitoring station, the software allows the processing and transmission of the fire information; and (3) the spatial mapping component, which integrates the information on forest distribution according to fire risk. This has a summarizing function and allows for the managers to look up information on fires. In addition to information on fires reported via email and SMS, the software also incorporates a map of fire points that allows users to make direct visual observations remotely (Figure 11).

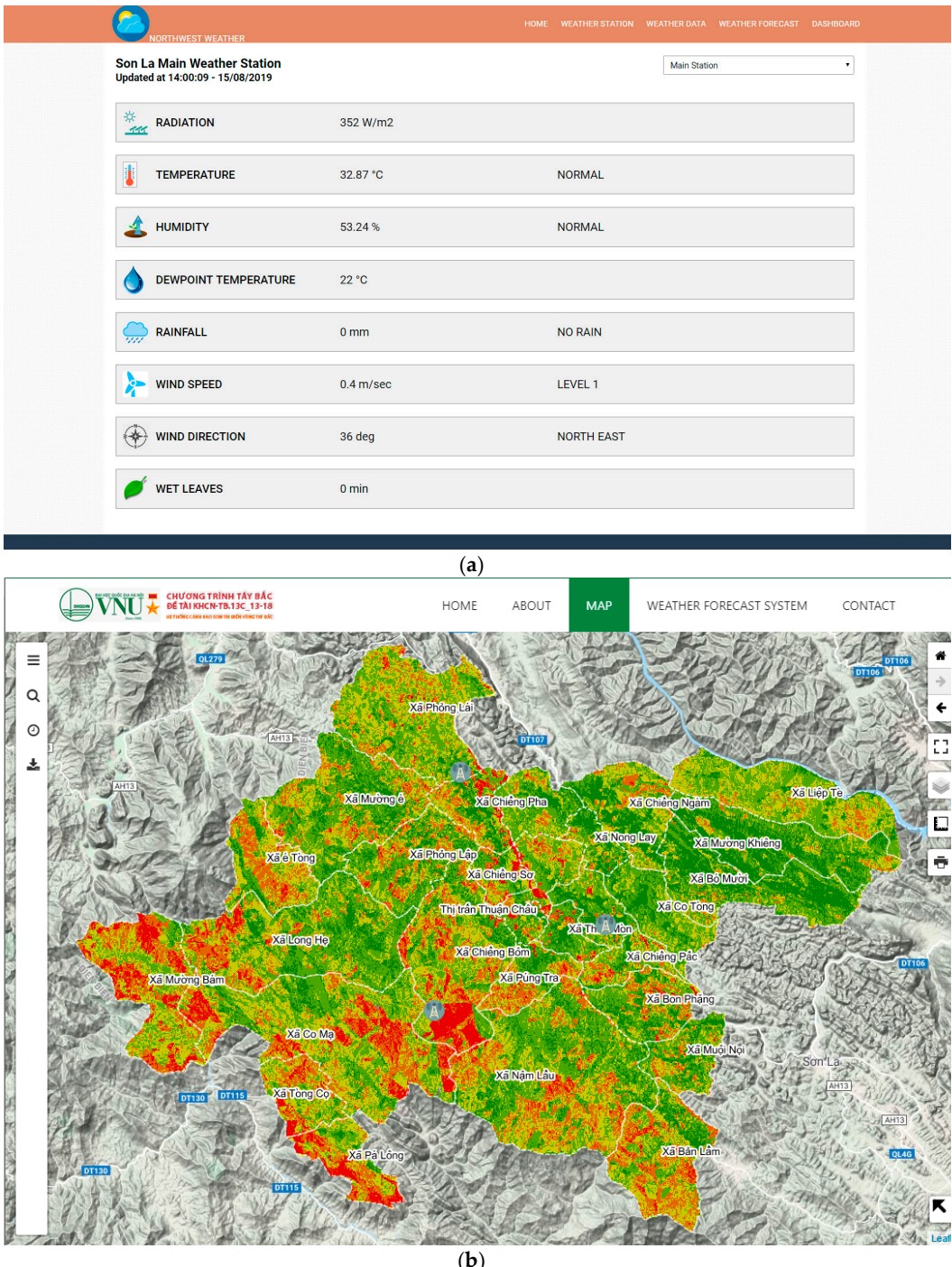

**Figure 11.** The user interface of the software (**a**) and the forest fire warning website (**b**).

## 5. Discussion and Conclusions

Currently, forest fires are one of the most serious challenges for forest development and protection efforts in Vietnam and efforts towards forest fire prevention have necessarily become increasingly focused [36]. The Forest Protection Department has several possible sources of forest fire information that can be collected relatively effectively (e.g., from local forest rangers, and from remote sensing stations). The use of remote sensing technology in forest fire monitoring by the Forest Protection Department mainly involves the use of thermal channels of low-resolution satellite images with a high frequency of observation. However, the development of more reliable information about forest fires

systems requires interdisciplinary cooperation between experts in a range of different disciplines. The research has integrated modern technologies including identifying areas of forest fire risk, the early warning of forest fires, monitoring forest fires (fire point system), damage assessment (in terms of fire area) and the dispersal of results (via WebGIS). The forest fire information system developed here has advantages compared to other systems. Specifically, the forest fire forecast function is improved by using and combining various meteorological data sources, including the meteorological data collected from ground points and the meteorological data from satellites. All the information about fires and fire forecasting is presented on an interactive map interface using uniform GIS technology.

Nevertheless, the use of these results in daily forest management and protection still faces some limitations due to delays in conveying the results due to constraints relating to satellite observation frequency and the transfer of results may be slow or lack specific information due to district administrative procedures. Notwithstanding such challenges, the results are of considerable practical value because the system integrates existing tools to provide information at each stage of risk assessment, warning, and fire point notification to damage assessment. Furthermore, this information is integrated into a system capable of providing fast information (web-based) and capable of searching and attributing data both spatially and temporally (due to possible storage and search functions over time).

The results show that the integration of GIS technology and multi-indicator analysis tools (MCA) with layers of information related to forest fire risk, such as humidity, wind direction, forest type, slope, slope direction, stream density, slope, distance to residence, main roads, trails, etc., allows for the construction of reliable fire risk maps. The comparison of model results with historical data confirms the accuracy and reliability of the system. The application of the semi-quantitative AHP method and the expert systems approach in the evaluation is shown to be a valuable additional element and this is especially helpful in developing a real-time early warning system. The results provide forest fire risk maps at the district and commune-level administrative unit scale to enable effective forest fire prevention in Son La.

In addition, the results of the project contribute to an understanding of how the science and technology applications of remote sensing and GIS can be integrated with expert knowledge to improve forest management. Rapid development of freely available remote sensing data with high resolution and repeatability has resulted in its adoption towards a wide range of applications, including forest fire monitoring and prevention. There are many opportunities to use remote sensing data in the management and monitoring of natural resources, including forest resources. As is demonstrated here for Son La province, the proactive scientific application of remote sensing technology and GIS contributes to significant efficiency improvements in forest fire prevention and firefighting in terms of time and cost.

**Author Contributions:** Conceptualization, P.X.C.; data curation, Q.H.N. and X.L.N.; formal analysis, P.X.C. and D.N.B.T.; funding acquisition, T.Y.C.; investigation, T.V.H. and Y.M.F.; methodology, N.T.N.; project administration, N.T.N.; resources, D.N.B.T.; software, Q.H.N.; supervision, T.Y.C. and M.E.M.; validation, Y.M.F.; visualization, X.L.N.; writing—original draft, T.V.H., Q.H.N. and D.N.B.T.; writing—review and editing, T.V.H., T.Y.C., and M.E.M. All authors have read and agreed to the published version of the manuscript.

**Funding:** This research received no external funding.

**Acknowledgments:** This article is the result of a state-level project titled "Research on modeling and system of sub-regional weather forecasting and warning of flood, forest fire and agricultural pests at district level in the North West Vietnam", Code: KHCN-TB.13C/13 -18 and has been financed by the National Program for Tay Bac, VNU Hanoi, Viet Nam; supervised and assisted by Geographic Information Systems Research Center, Feng Chia University, Taiwan.

**Conflicts of Interest:** The authors declare no conflict of interest.

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
