# Peer review of "Mapping Forest Fire Risk and Development of Early Warning System for NW Vietnam Using AHP and MCA/GIS Methods"

_applsci, doi:10.3390/app10124348_

Round 1
Reviewer 1 Report
Authors have corrected all previous remarks. Paper can be published in a present form.
Author Response
Thank you for supporting us during revision process of our paper.
Regards.

Reviewer 2 Report
No comments. Concerns in previous review have been met.
Author Response
Thank you so much for your positive comments.

This manuscript is a resubmission of an earlier submission. The following is a list of the peer review reports and author responses from that submission.
Round 1
Reviewer 1 Report
The authors follow the recommendations suggested in the last review.
So, I think that only spelling should be checked.
Regards
Author Response
Response: Dear Reviewer 1, we have received your comments. Thank you so much during this time, your positive and detailed comments helped our paper improve a lot.
We did check English spell. Thank you for your kind support!
Wish you all the best!
Kind regards,
Van

Reviewer 2 Report
Authors have corrected all previous remarks. Manuscript can be accepted in a present form.
Author Response
Response: Dear Reviewer, we have received your comments. Thank you so much during this process, your positive and detailed comments helped our paper improve a lots.
Thank you for your kind support!
Wish you all the best!
Kind regards,
Van

Reviewer 3 Report
Overall good job.
With reference to line 303, please elaborate on how distance from residential points was determined or calculated.
With line 118, please elaborate on how different forest types relate to fuel burns. Mention some dominant species, for example. For example, pine and acacia are mentioned in line 209.
In the conclusions, can a statement be made about the effectiveness of this model compared to other models? Can the model be applied to other parts of the world?
I have marked some suggestions directly on the manuscript that will improve the clarity of the writing. Also, some other minor points to consider.

Author Response
Response Reviewer: Thank you very much for your comments. Those comments are all valuable and very helpful for revising and improving our paper. We have studied comments carefully and have made correction which we hope meet with approval. The main corrections in the paper and the responses to the comments are as flowing:
Point 1: With reference to line 303, please elaborate on how distance from residential points was determined or calculated.
Response 1: Yes, We extracted the residential area from Land Use Change map. After that, the Euclidean Distance method was applied to calculate the distance to the residential field.
Point 2: With line 118, please elaborate on how different forest types relate to fuel burns. Mention some dominant species, for example. For example, pine and acacia are mentioned in line 209.
Response 2: Yes, Based on previous research, each forest type will have a different burned ratio. Our study area has various types of forests, so we divided them into 5 assessment groups based on a potential burned ratio. For example, pine and acacia are shrub species in the planted forest (assessment level is 5 - highest), which has a high potential burned ratio. Or bamboo and mixed forest are also in the assessment level 5 group.
Point 3: In the conclusions, can a statement be made about the effectiveness of this model compared to other models? Can the model be applied to other parts of the world?
Response 3: yes, we have a statement (Line 468-470) to talk about this issue. This model would be applied for other parts in the world, however, each region is different from geographical and climate, therefore whether or not this model should apply for that area, it depends on the government and local there.
Point 4: I have marked some suggestions directly on the manuscript that will improve the clarity of the writing. Also, some other minor points to consider.
Response 4: Yes, we have revised follow the attachment file you note for us, and we noted for yellow highlight. Thank you so much to correct english for us. I appreciate so much for your support! Wish you all the best!
Best regards,
Van
